# New Insights into Hop Latent Viroid Detection, Infectivity, Host Range, and Transmission

**DOI:** 10.3390/v16010030

**Published:** 2023-12-23

**Authors:** Osama O. Atallah, Sherin M. Yassin, Jeanmarie Verchot

**Affiliations:** Department of Plant Pathology & Microbiology, Texas A&M University, College Station, TX 77843, USA; osama3@tamu.edu (O.O.A.); sherin.abouzaid@ag.tamu.edu (S.M.Y.)

**Keywords:** viroid, hemp, Pospiviroidae, diagnostics, host range, phylogeny

## Abstract

Hop latent viroid (HLVd), a subviral pathogen from the family *Pospiviroidae*, is a major threat to the global cannabis industry and is the causative agent for “dudding disease”. Infected plants can often be asymptomatic for a period of growth and then develop symptoms such as malformed and yellowing leaves, as well as stunted growth. During flowering, HLVd-infected plants show reduced levels of valuable metabolites. This study was undertaken to expand our basic knowledge of HLVd infectivity, transmission, and host range. HLVd-specific primers were used for RT-PCR detection in plant samples and were able to detect HLVd in as little as 5 picograms of total RNA. A survey of hemp samples obtained from a diseased production system proved sole infection of HLVd (72%) with no coexistence of hop stunt viroid. HLVd was infectious through successive passage assays using a crude sap or total RNA extract derived from infected hemp. HLVd was also highly transmissible through hemp seeds at rates of 58 to 80%. Host range assays revealed new hosts for HLVd: tomato, cucumber, chrysanthemum, *Nicotiana benthamiana*, and *Arabidopsis thaliana* (Col-0). Sequence analysis of 77 isolates revealed only 3 parsimony-informative sites, while 10 sites were detected among all HLVd isolates available in the GenBank. The phylogenetic relationship among HLVd isolates allowed for inferring two major clades based on the genetic distance. Our findings facilitate further studies on host–viroid interaction and viroid management.

## 1. Introduction

Viroids are pathogenic circular RNAs that infect members of the kingdom *Plantae* and are among the economically significant phytopathogens for numerous species of herbaceous and woody plants [1,2]. Viroids are generally highly contagious and can be transmitted mechanically by farm tools, insects, pollen, seeds, parasitic plants, shoot/root grafting, and recently fungal phytopathogens [1,3,4,5,6]. Notably, viroids such as citrus exocortis viroid (CEVd) have a wide host range, infecting plants from different botanical groups like trees and herbaceous plants [1,7]. Viroids are capsid-less genomes of small covalently closed circular RNA molecules with no encoded proteins [8,9,10,11]. Similar to viral pathogens, viroids enter the cell and rely on host factors/enzymes to achieve successful infection. 

Viroid replication occurs in either the nucleus or chloroplast, which also contributes to defining their taxonomic assignments to families *Pospiviroidae* and *Avsunviroidae,* respectively [11]. *Pospiviroidae*, named for *Potato spindle tuber viroid*, encompasses the major known viroid species whose genomes assume a rod-like or quasi-rod-like conformation with a central conserved region (CCR) and terminal conserved hairpins (TCH) or regions (TCR) [1,12,13]. *Pospiviroidae* includes the nuclear-replicating genera *Apscaviroid*, *Cocadviroid*, *Coleviroid*, *Hostuviroid*, *and Pospiviroid* [14]. These viroid genomes contain specific signals directing the viroid RNA into the nucleus where the host DNA-dependent RNA polymerase II drives its replication via an asymmetric RNA-to-RNA rolling circle mechanism, producing oligomeric RNA intermediates [1,11,15,16,17,18,19,20]. A host RNAse III cleaves the RNA oligomers, which are then circularized by a host DNA ligase I acting on the free termini [1,21,22,23]. Cellular mechanisms underlying disease and pathogenesis are poorly understood, although a few studies suggest that viroid RNAs may bind to host mRNAs or proteins and prevent their functions [13,24,25]. Genome-wide analysis of tobacco (*Nicotiana tabacum*), *N. benthamiana*, and commercial hop (*Humulus lupulus* L.) inoculated with five different viroids indicates that different viroids influence individual mediator (MED) subunits in a manner that may stimulate host transcriptional reprogramming. MED is a multi-subunit protein complex that regulates plant gene expression through interactions with RNA polymerase II and transcription factors [26,27]. Emerging evidence indicates that viroid small RNAs influence (post) transcriptional gene silencing and DNA methylation [11,28,29].

Our research focuses on hop latent viroid (HLVd), which belongs to the genus *Cocadviroid* along with three other viroids: citrus bark cracking viroid (CBCVd), coconut cadang-cadang viroid, and coconut tinangaja viroid [30]. Most members of the genus *Cocadviroid* have genomes extending between 246 and 284 nt, and the HLVd genome is 256 nt [12].

HLVd was first isolated from commercial hop plants in Spain by Pallas et al. (1987), and its biological and molecular features were further described one year later by Puchta et al. (1988) [12,31]. HLVd does not often cause a visible disease phenotype in hop plants, hence it has been termed a latent viroid [12,32]. HLVd exists worldwide in many hop-growing regions in Belgium, Brazil, China, Czech Republic, Germany, Japan, Poland, Slovenia, Turkey, the UK, and the USA [3,12,33,34,35,36,37,38,39,40,41]. In addition to commercial hop plants, scientists reported HLVd causes leaf distortion and yellowing in Japanese hop (*Humulus japonicus* Sieb. and Zucc.) and stinging nettle (*Urtica dioica* L.) [12,42]. Only mechanical and low-level (6%) seed transmission was proven for HLVd in hop plants [12,43]. Recently, growers have reported devastating economic losses due to “dudding disease” caused by HLVd in industrial hemp (*Cannabis sativa* L.) [38,40]. In mature hop and hemp cones, the secondary metabolite contents are greatly affected, and the cone yield is significantly reduced [3,44,45]. The contents of alpha bitter acid, essential oils, prenylated flavonoids, tetrahydrocannabinol, and terpenes can greatly fluctuate, which negatively impacts the total yield quantity and value. Losses to the hemp industry were estimated to be four billion dollars per year [3]. Two herbaceous plants outside this ecological niche were reported to be susceptible to HLVd infection after heat treatment: tomato (*Solanum lycopersicum*) and *N. benthamiana* [46].

Because of its latency, HLVd has no apparent symptoms in young hemp and hop plants and can go undetected. Until now, biological indexing of young susceptible plants has not yet been routinely implemented. Industrial hemp plants are often clonally propagated, and there is a need for new methods to assess their phytosanitary status before vegetative propagation. Among the original techniques for viroid detection and diagnosis include direct polyacrylamide gel electrophoresis (PAGE) and bidirectional (return) electrophoresis [12,47,48,49]. Molecular hybridization methods including Northern, dot-, and tissue-blotting were among the original identification methods [12,49]. In the past decades, RT-PCR and RT-qPCR technologies are more often employed due to their fast, accurate, and reliable results [49,50,51,52]. The single-strand conformation polymorphism technology detects variations in the 2-D genome structure among viroid species [53,54]. The newest studies employ loop-mediated isothermal amplification (RT-LAMP) and commercial proprietary methods (e.g., MFDetect™ by MyFloraDNA in Sacramento, CA, USA; AmplifyRP^®^XRT by Agdia in Elkhart, IN, USA; and TUMIGlow™ by TUMI Genomics in Fort Collins, CO, USA) to detect HLVd and other viroids in leaf and root tissues [55,56,57,58].

Several peer-reviewed studies confirming HLVd in industrial hemp used RT-PCR and the primer pair HLVdF and HLVdR designed by Matousek and Patzak (2000) [33,38,40,41,43,57], but the assay sensitivity using RNA extracted from various hosts has not been reported. In addition, the host range of HLVd has not been studied until now and our understanding of its narrow host range is mainly anecdotal. We report sensitive RT-PCR detection of HLVd using primers designed to detect HLVd RNA for simple interpretation of PCR products by agarose gel electrophoresis. We inoculated plants belonging to six species that are known hosts to several viroids with HLVd-containing sap or RNA preparations from HLVd-infected hemp plants, revealing a wider range of plant species than previously reported. We confirmed vertical transmission through seeds as a significant means of disease dispersal.

## 2. Materials and Methods

### 2.1. Plant Materials and Infected Samples

Thirteen hemp samples (leaves, petioles) and 20 g of hemp seeds from an HLVd-contaminated nursery were kindly provided by Dr. Tassa Saldi (TUMI Genomics), and many tested positive for HLVd at the TUMI Genomics facility, Fort Collins, Colorado, USA. These samples were shipped in a sealed package with ice packs and kept at −80 °C until further analysis. Leaf and petiole samples were used as a source of inoculum after confirming HLVd-positive status by RT-PCR and Sanger sequencing. The HLVd inoculum was propagated and maintained on hemp plants (cv. ‘Cherry Wine’), kindly provided by Dr. Russell W. Jessup, Soil and Crop Sciences at Texas A&M University, College Station, TX, USA. Potential hosts/reservoirs for HLVd included the following: *Arabidopsis thaliana* (Col-0), *N. benthamiana*, *Solanum lycopersicum* (tomato cv. ‘Moneymaker’), *Cucumis sativus* (cucumber cv. ‘Harris’), and *Chrysanthemum morifolium* (chrysanthemum var. ‘Yellow Fall’). Plants were grown in 20 cm pots filled with Jolly Gardener^®^ Pro-line C/25 growing mix amended with Osmocote^®^ slow-release fertilizer (20:20:20) and placed in a growth chamber at 23 ± 2 °C with 12 h light/dark.

### 2.2. RNA Extraction, RT-PCR, and Primer Sensitivity

Freshly collected leaf and root samples were thoroughly washed under running tap water, rinsed three times with sterile ddH_2_O, and air-dried on filter paper. Total RNA was extracted from leaves and roots using 150 mg of tissue frozen in liquid nitrogen, a mortar and pestle for grinding, and the Maxwell^®^ 16 LEV SimplyRNA Tissue kit (Promega Corp., Madison, WI, USA). RNA was treated with TURBO^®^ DNAse I. The integrity and concentration of the total RNA extracts were analyzed using the NanoDrop™ 2000 spectrophotometer (Thermo Fisher Scientific, Waltham, MA, USA).

Total RNA (0.5 µg), random hexamers, and Maxima™ reverse transcriptase were used to synthesize cDNA (ThermoFisher Scientific Corp, USA). GoTaq^®^ polymerase (Promega™ Corp.), HLVd-specific primers and HSVd-specific primers (0.5 µM final concentration) for viroid targets or 7SL RNA [58,59,60] (Table 1), and 1 µL of cDNA were used to conduct endpoint PCR. We designed primers UCCR-F and UCCR-R, which anneal to the upper strand of the HLVd genome’s central conserved region (CCR) (Tabe 1). These primers are similar to ones used by Matousek and Patzak [43] with slight modifications. UCCR-F extends from nt positions 51 to 74 while UCCR-R extends from nt positions 35 to 58 of the reference HLVd complete genome (GenBank ID: NC_003611.1). The UCCR-F/R primers overlap by eight nts which naturally include *Bam*HI restriction sites (Figure 1A).

Primers were optimized using Primer3 software v. 4.1.0 (https://primer3.ut.ee/). PCRs were performed by initial denaturation at 94 °C for 3 min followed by 35 cycles of denaturation at 94 °C for 20 s, annealing at 58 °C for 20 s, and elongation at 72 °C for 20 s. The final extension was at 72 °C for 10 min. PCR amplicons were examined on 2% TBE agarose gels stained with ethidium bromide and photographed with a ChemiDoc™ MP imaging system (Bio-Rad Laboratories, Inc., Hercules, CA, USA). PCR amplicons were purified using the Gel/PCR DNA fragment extraction kit (IBI Scientific, Peosta, IA, USA) and sent to Eton Bioscience Inc. (San Diego, CA, USA) for sequencing.

To determine the sensitivity of the PCR primers for nucleic acid detection, ten-fold serial dilutions of RNA and plasmids containing HLVd cDNA cloned inserts were suspended in salmon sperm DNA and nuclease-free water. The construction of a plasmid containing HLVd cDNA is detailed below. For HSVd detection, we used F1 and R1 primers (Table 1) developed by Bernad and Duran-Vila [58,59]. Although this pair of primers was designed to detect HSVd in citrus, they also detected HSVd isolates known to infect hops as shown by multiple sequence alignment (Appendix A). The accession numbers HE575344, HE575345, HE575346, HE575347, and HE575348 refer to HSVd recovered from hop plants in Slovenia [59].

### 2.3. Viroid Inoculum and Plant Inoculation

Crude sap and total RNA were prepared from 10 g of HLVd-infected hemp and healthy control leaves. First, the midribs were cut and discarded, and then leaf tissues were ground in 10 mL of ice-cold phosphate buffer (pH 7.2) with 1% (*v*/*v*) 2-mercaptoethanol using a prechilled mortar and pestle placed in ice. Leaf extracts were filtered with double-layered cheesecloth and diluted 10-fold in the same buffer plus 0.1% (*v*/*v*) Silwet™ L-77 (PhytoTech Labs, Lenexa, KS, USA). Sap was again tested for HLVd by RT-PCR. Plants were inoculated with 0.5 mL of the freshly prepared sap at the rate of 25 µL per hemp leaflet after light dusting with carborundum 600 grit. Undivided leaf blades were inoculated with 125 µL per leaf and four true leaves were rub-inoculated. Inoculated plants were misted with sterile water until run-off and kept in the shade overnight before placing them in the growth chamber.

Total RNA inoculum was prepared from filtered sap using TEMS buffer (0.1 M Tris-HCl, 0.01 M EDTA,0.1 M NaCl, and 0.01% 2-mercaptoethanol, pH 7.2) and 2M LiCl. After assessing RNA quality and concentration, RT-PCRs were performed to confirm the presence of HLVd in the samples. One µg (1 ng/1 µL) of total RNA was first supplemented with 0.1% (*v*/*v*) Silwet™ L-77 and then applied to each plant.

Positive control constructs were prepared by cloning the HLVd sequence (GenBank ID: OR338705) by RT-PCR amplification from an infected plant in the pGEM-T Easy vector system under the T7 promoter. These pGEM clones containing mono- and dimeric HLVd sequences were verified by Sanger’s sequencing (Eton Bioscience Services, San Diego, CA, USA).

### 2.4. Seed Transmission Assay

Seeds from infected hemp plants were kindly provided by Dr. Tassa Saldi (TUMI Genomics). Seeds were rinsed in sterile ddH_2_O for 5 min, 10% bleach for 2 min, and three times in sterile ddH_2_O for 5 min, then air-dried on filter papers in a laminar flow hood. Seeds were sown and then 100 seedlings with 4–6 mature leaves from each replicate were tested for HLVd by RT-PCR. A total of 100 seedlings from healthy seeds were used as a control.

### 2.5. Sequence Analysis and Phylogeny

Gel-purified PCR products were sequenced by Eton Biosciences (Research Triangle Park, NC, USA) in the forward and reverse directions. Sequence reads were manually curated. HLVd genomes were assembled and provided to NCBI GenBank to obtain accessions (Appendix A). CLUSTAL OMEGA in Geneious Prime^®^ (v 2023.0.4) (Biomatters Ltd., Aukland, New Zealand) was used to perform multiple sequence alignments. DnaSP 6 (v 6.12.03) was used to assess nt polymorphisms, diversity, and parsimony-informative sites among HLVd sequences [61], with a 10 nt window and 10 nt step size. Maximum Likelihood [62] in MEGA X [63] was used to infer the phylogenetic relationships among the HLVd sequences and fully curated genomes in the NCBI database (Appendix A). HSVd (NC_001351.1) and CBCVd (NC_003539.1) were outgroups. The best substitution model predicted in MEGA X was the Kimura 2-parameter (K2) plus gamma distribution. To assess branch support, datasets were sampled with 500 bootstrap replicates, where values equal to or greater than 70% were considered significant. The tree was re-rooted in FigTree v1.4.4 (http://tree.bio.ed.ac.uk/software/figtree/; accessed 7 December 2023).

### 2.6. Imaging and Data Analysis

Images were obtained using a Nikon D3400 digital camera with a Nikkor 18–55 mm zoom lens and assembled into figures using Adobe^®^ Photoshop. GraphPad Prism (v8.3.0; GraphPad Software, La Jolla, CA, USA) was used for one-way analysis of variance (ANOVA) tests and for generating plots. For post hoc analysis, Tukey’s honestly significant difference (HSD) test for pairwise comparison among treatments was used (*p* = 0.05).

## 3. Results

### 3.1. Sensitive RT-PCR Diagnostic Detection of HLVd in Hemp Plants

HLVd RT-PCR products were detected by gel electrophoresis when using between 500 ng/μL and 0.5 pg/μL RNA extracted from infected asymptomatic plants (Figure 1B,C). There was no product amplification when using UltraPure water rather than RNA in the two-step RT-PCR (Figure 1C). PCR amplicons were also detected by gel electrophoresis using 2 fg/µL, 200 ag/µL, 20 ag/µL, and 2 ag/µL of HLVd-encoding plasmids, implying that this assay is extremely sensitive (Figure 1C).

To assess the possibility of HLVd self-priming in the reverse transcription (RT) reactions, we performed RT with and without primers or RNA in the RT reactions and then compared PCR amplicons obtained from these reactions [64,65,66]. Amplicons were detected by gel electrophoresis only in samples in which the RT reaction had RNA and primers (top panel, Appendix A). To further assess the possibility of false priming at the PCR level, the same RT reaction products were used in PCR reactions with and without the amplification primers (UCCR-F/R). PCR amplicons were detected by gel electrophoresis only when the primers were present (bottom panel, Appendix A).

### 3.2. Viroid Survey and Genetic Variation

We performed diagnostic RT-PCR tests using hemp samples (*n* = 111) provided by a producer in Colorado and the UCCR-F/R primer set. Eighty samples tested positive for HLVd (Figure 1D and Appendix A). Though hop stunt viroid (HSVd) is not a common concern for hemp growers in the USA, we performed RT-PCR using an HSVd primer set (Table 1) described by Bernad and Duran-Vila [58]. We found no evidence of these two viroids co-existing in this sample population (Figure 1D).

To analyze the genetic diversity across HLVd in these samples, we confirmed 41 full-length sequences from this sample population (Appendix A), which we aligned (along with the NCBI reference genome) using CLUSTALW (Appendix A). The average pairwise differences across the lengths of the sequences (*Pi*) ranged from 0.00488 to 0.0378, from which a small variance among isolates can be inferred. Three informative changes were observed among the HLVd sequences recovered in this study: A_158_G, A_196_T, and C_223_T (Figure 2A). Next, we searched NCBI GenBank for complete HLVd sequences and identified 21 complete genomes to incorporate in this study (Appendix A). Nt diversity was again analyzed using the 71 isolates, and *Pi* values ranged from 0.00282 to 0.05545 (Figure 2B). The A_158_, A_196_, and C_223_ positions were statistically significant and the final nucleotide position 256 was quite variable across isolates. Tajima’s test also identified one nucleotide position as significant among the 71 isolates: 196 with *D* = 0.87 (Figure 2C).

Next, maximum likelihood (ML) trees were constructed and rooted with two other members of the genus *Cocadviroid*, HSVd (NC_001351.1) and CBCVd (NC_003539.1) (Figure 3). The branch patterns were examined to assess how the three nt positions identified by *Pi* and *D* values might influence the phylogenetic relationships among isolates. The tree bifurcated into major clusters defined by either an A or T at position 196. Among the isolates with A_196_, there was a subtree with G replacing A at position 158 and another cluster with T replacing A at position 237. Among the T_196_ group, there were only two isolates with the C_223_T mutation including the isolate OR338707 which was at the base and the isolate OR338743 near the top of this group. Among the T_196_ group are two isolates defined by C_128_T mutation, two isolates with T_200_C and A_234_G mutation, and two isolates with AAT replacing CCC at positions 164 to 166.

### 3.3. Infectivity of HLVd Inoculum and Its Distribution in Mature Hemp Plants

To determine the efficiency of HLVd inoculum types, crude sap and RNA extract from an infected hemp plant were used to inoculate healthy hemp seedlings. Sap and extracted RNA from healthy hemp plants were used as inoculum controls. Four leaves of 8-week-old plants were inoculated with 500 µL of contagious sap or 1 µg of RNA per plant. An amount of 10 plants was used per treatment, and the whole trial was repeated five times, providing a total of 50 plants per treatment. Samples of newly emerging non-inoculated leaves and roots were collected at 3 weeks post inoculation (wpi) and tested by RT-PCR. We saw that 100% of the sap-inoculated plants were infected, while 60% of the RNA-inoculated plants were infected (Figure 4A,B). Plants testing positive in the leaves also tested positive in the roots. There was only one plant inoculated using RNA whose roots and not leaves tested positive for HLVd (Figure 4A). Plants were asymptomatic when sampled (Appendix A).

We tracked the pattern of HLVd spreading in infected hemp plants following inoculation of a single leaflet of a palmate leaf with contagious or healthy sap (*n* = 10 plants). At 0, 3, 7, and 14 dpi, inoculated leaves, upper leaves, and root samples were subjected to surface sterilization through 5 min of rinsing with sterile water and 1 min in 5% household bleach, rinsing three times with sterile water for 2 min, and air-drying on filter papers in a laminar air flow cabin. RNA was extracted from leaves and roots and RT-PCR was performed to detect HLVd sequences. HLVd was first detected at 3 dpi by RT-PCR in 100% of the surrounding leaflets of the inoculated palmate leaves (Figure 5A) and continued to test positive at 7 and 14 dpi. HLVd infection appeared in the roots in 10% of plants at 3 dpi, 80% at 7 dpi, and 100% at 14 dpi. HLVd was detected in the leaf directly above the inoculated leaf at 3 dpi and then in the terminal upper leaves at 14 dpi (Figure 5B).

### 3.4. HLVd Host Range and Disease Symptoms

*Arabidopsis thaliana* (Col-0), *Nicotiana benthamiana*, tomato, cucumber, and chrysanthemum plants were inoculated with HLVd contagious sap. Crude sap from healthy hemp plants was used for mock inoculation. At 3 wpi, RNA was extracted from the inoculated leaves, upper non-inoculated leaves, and roots. HLVd was in 100% of the inoculated plant leaves, except for chrysanthemum where only 60% of plants became infected locally (Figure 6, Table 2). HLVd was detected in 40% to 100% of the roots and upper leaves of the inoculated plants, except in *N. benthamiana* plants where HLVd was detected in the roots but not the upper non-inoculated leaves (Table 2). No HLVd sequences were detected in the mock-treated plants. HLVd sequences recovered from five different tomato and *N. benthamiana* plants were sequenced and compared to the original sequence used for inoculation by multiple sequence alignment. The progeny HLVd sequences were identical to the parental sequence and no single nt change was observed.

The HLVd-inoculated plants were monitored for phenotypic alterations for two months after inoculation, when the majority of plants begin to bloom. No visible symptoms were detected on infected tomatoes, chrysanthemum, cucumber, or *N. benthamiana* plants. We inoculated Arabidopsis plants with 10 to 11 rosette leaves (growth stage 1.10 to 1.11) and these developed die-back symptoms, starting from the older leaves and gradually going inwards until the whole plant died (Figure 7A). Infected plants did not develop die-back at the same time, as some started to die as early as 14 dpi while others reached complete decline and tissue collapse by 19 dpi which is just before starting the bolting stage. During the same 19 d period, mock-inoculated plants continued to grow normally and reached the growth stage number 6.00 where the first flower is open (Figure 7A). This experiment was repeated using older Arabidopsis plants with 14 rosette leaves (growth stage 1.14) and the die-back started to develop after bolting at stage number 6.90. The inflorescences became dry and eventually died (Figure 7B). No abnormalities were observed on seeds or siliques collected from infected or mock-treated plants (Figure 7C,D). The shapes of siliques were similar between infected and mock-inoculated plants. The seed yield was between 30 to 50 seeds per silique, depending on the silique’s length, in both infected and mock treatments.

To substantiate the observation that *A. thaliana* is a host for HLVd, we inoculated plants at the same growth stage (1.10 to 1.11) with infectious HLVd dimeric transcripts synthesized from pGEM plasmids using T7 RNA polymerase. Plants were each inoculated with 200 ng transcripts after dusting with carborundum 600 grit. Symptoms of decline appeared after 14 dpi (Figure 7E). As seen using sap inoculum, decline initiated at the tips of multiple leaves and gradually rendered the canopy dead. Diseased plants were sampled at 14 dpi by collecting the youngest green tissues and infection was confirmed by RT-PCR (Figure 7F).

### 3.5. Vertical Transmission of HLVd via Hemp Seeds

To test whether HLVd can be transmitted through generations via seeds of hemp, we ran two experiments in which we crossed three HLVd-infected females with a pollen from healthy male (T1) and three healthy females with a pollen from HLVd-infected male (T2). HLVd infection in mother plants and pollen was confirmed by RT-qPCR at TUMI Genomics. Healthy mother plants and pollen were also confirmed as free of HLVd infection using the same technique. Each experiment was replicated thrice. After successful fertilization and seed maturity, the seed yield was collected, surface sterilized, and sown in pots. The whole procedure is summarized in (Figure 8A). HLVd was detected in plants infected through the seeds in both treatments 1 and 2 (Figure 8B). In treatment 1, 84.3% of 300 plants were HLVd positive, and in the other treatment (2) 58% of another 300 plants were positive. The percentages of plants infected through seeds in T1 and T2 were statistically insignificant. No HLVd-positive samples were detected in the 100 control plants.

## 4. Discussion

The global trade in crop seeds and seedlings has grown in volume since the 1930s and this has contributed to the trend of emerging viruses and viroids, and HLVd is not an exception [67,68,69]. HLVd is a major threat for the hop (*Hamulus lupulus*) and hemp (*Cannabis sativa*) industries worldwide [12]. HLVd is a highly transmissible pathogen that can be easily transmitted via mechanical means. Early and accurate diagnosis is an essential step in disease management. However, HLVd infection usually remains latent, which makes it challenging for growers to diagnose the disease at earlier stages of infection. Viroids are made of RNA only, which limits the use of detection techniques exclusively to nucleic-acid-based methods only. Therefore, we established this research to expand our knowledge on HLVd detection, diagnosis, host range, and transmission so we can increase the efficiency of HLVd management protocols.

HLVd was first described in 1988 [12,31] using PAGE electrophoresis, HPLC, blotting, and nucleic acid hybridization techniques. A decade after this, end-point PCR was employed for HLVd detection using viroid-specific primers and has since become a popular and feasible tool for the accurate detection and quantification of viroid loads in infected tissues [38,40,43,70,71,72,73]. Biological indexing is no longer beneficial for discriminating between infected and healthy hemp plants due to a lack of characteristic symptoms of novel versus existing viruses and viroids. Nowadays, RT-PCR has become a commercially available technology for HLVd detection via many third-party laboratories in the United States. Our current study starts to shed light on the role of primers in sensitive diagnostic detection of HLVd by RT-PCR. We detected HLVd in minute quantities of total RNA samples (5 pg/µL) extracted from different parts of the plant, as well as viroid cDNAs (2 ag/µL) cloned into a bacterial plasmid. We also demonstrated that there was no self-priming activity of the viroid sequences through the RT-PCR amplification. Our results demonstrate the high accuracy and reliability of RT-PCR assay in HLVd detection in infected hemp samples.

HLVd is among several viral and subviral agents that attack hemp and cause several types of losses in product quality and total yield, such as HSVd, hemp streak virus, hemp mosaic virus, cannabis cryptic virus, and beet curly top virus [3,70,73,74]. Disease symptoms can be significantly affected by pathogen population within the same niche, and synergism or superinfection exclusions are among the widely known types of interactions among viral pathogens within plants [75,76,77,78]. We studied the possibility of co-existence between HLVd and HSVd, two of the most widely spread viroids, in hemp and [77] found no evidence of HSVd sequences in this large plant population (*n* = 111). In addition, HSVd was not detected in HLVd-free hemp plants, suggesting that HSVd is absent from this plant nursery.

RNA sequence analysis and phylogeny of HLVd genomes recovered from field samples and NCBI GenBank indicated slight but discriminative variations. Three key nucleotide changes, A_158_G, A_196_T, and C_223_T, were identified among all tested HLVd isolates. The A or T at position 196 was key for differentiating two major groups of HLVd isolates. The A_158_G change existed in a subset of six newly sequenced isolates within the A_196_ group. The C_223_T change appears in one isolate at the base of the T_196_ group and a single isolate at a more recent branch at the top of the tree. These three mutations reside in the loop regions of the genomic RNA and it is worth speculating that they might influence the stability of the secondary structures or interactions with cellular factors that bind to these loop regions. Triple and quadruple mutations were previously observed in HLVd isolates upon heat treatment for hemp mericlones [46,79,80]. The same study inferred the stability of the upper central conserved region while the pathogenicity domain was a hotspot for accumulating mutations [80]. They added that those mutations destabilize the secondary structures within the affected sequences, which has also been demonstrated for potato spindle tuber viroid (PSTVd) [81]. Matousek et al. (2003) [46] reported that heat-induced mutations in the HLVd genome enabled host range expansion. In 2003, researchers did not consider hemp as an alternative host to hops but used solanaceous hosts as test plants, which are typically used by plant virologists. In general, viroid transmission to Arabidopsis has not been previously demonstrated, although Daròs and Flores (2004) indicated that Arabidopsis has the necessary machinery to support replication of pospiviroid species and showed that positive and minus strand RNAs accumulate in transgenic plants expressing six different viroid species [82]. Future experiments will be performed to investigate the influence of adaptive mutations in the movement of HLVd from hops to hemp, as well as from hemp into Arabidopsis.

The inoculum of HLVd is infectious in several forms. It has been previously demonstrated that either crude sap, total RNA extract, or cDNA transcripts of HLVd are infectious and can successfully aid viroid replication within hemp [12,46,80]. Here we compared the infectivity of crude contagious sap versus total RNA extracts from infected hemp. HLVd sequences were detected in the roots and upper non-inoculated leaves of inoculated plants, which indicates successful movement of the viroid within the inoculated plants. Our findings added that crude sap is more effective than total RNA extract, which could be attributed to inoculum stability within plant sap due to the presence of natural buffering and stability factors [80,83]. A similar effect of contagious sap versus in vitro synthesized inoculum was observed by [84].

We further studied the spatial and temporal distribution of HLVd in infected hemp and we monitored the avenue and timing of invading new organs. HLVd propagates and proceeds to adjacent tissues that share the same petiole within the first 3 days of inoculation. After 7 days, HLVd reaches roots of inoculated plants to highly detectable levels. After 7 more days of inoculation, HLVd sequence reached the upper leaves of inoculated plants and successfully achieves movement through the systemic route. This route of systemic movement is common for viral and subviral agents infecting host plants, as plant viruses and viroids were detected in roots [85,86]. Knowledge of the timing of appearance in each part of infected plants can help with sampling the right tissue for detection.

Although pathogenicity of HLVd in hemp had been previously demonstrated by several researchers, the host range of HLVd is understudied. Besides hemp, it has been shown that HLVd thermomutants infect solanaceous plants [46]. In this study, we demonstrated that HLVd has expanded its host range beyond its natural host and invades other plants from unrelated families. Tomatoes, chrysanthemum, cucumber, *N. benthamiana*, and Arabidopsis were successfully inoculated with wild-type inoculum of HLVd with an unmutated sequence, and the viroid was detected in all plant organs by RT-PCR. This finding will allow us to study the viroid–host interactions and viroid replication factors. It is not surprising that the HLVd host range includes plants from unrelated groups. Viroid host ranges in general can include unrelated host plants [1]. CEVd, CBCVd, HSVd, and PSTVd are examples with hosts ranges including herbaceous and woody plants [87,88]. HLVd infection can be latent without apparent symptoms. However, we observed die-back disease and tissue collapse in the inoculated Arabidopsis after 2–3 weeks post inoculation. Further studies are needed to decipher the mechanisms employed by HLVd to elicit this phenotype.

Seed transmission is one of the most vital means of transmission evolved by plant viruses and subviral agents to overwinter during the absence of growing plant hosts. This can be a significant cause for accidental introduction into farm production systems that are a long distance from the original source. In 2001, Matousek and Patzak [43] showed that HLVd has very low likelihood of seed transmission in hops, not exceeding 6%, from which they assume its insignificance. In 2008, the same group confirmed in hops plants that HLVd is not seed-transmissible due to the nuclease activity in pollen [89]. In 2021, they studied transmission of CBCVd, apple fruit crinkle viroid, and PSTVd via *N. benthamiana* seeds, and they confirmed that those viroids are nontransmissible through seeds as they get eradicated during pollen development [71]. Other studies confirmed transmission of numerous viroids via seeds and pollen including PSTVd [90,91,92,93,94], which supports our finding [3,6,91,93,94]. A high percentage (82%) of seed transmission rate for columnea latent viroid was also reported in eggplant [95]. Surprisingly, a high percentage of seed transmission for HLVd was detected in our current study. Transmission percentages of 58 to 84% were observed when an infected male parent or infected female parent was used, respectively. Future studies will be performed to investigate the potential for pollen and seed transmission of HLVd across host species and to develop methods for screening hemp seeds to segregate and certify healthy seeds.

## Figures and Tables

**Figure 1 viruses-16-00030-f001:**
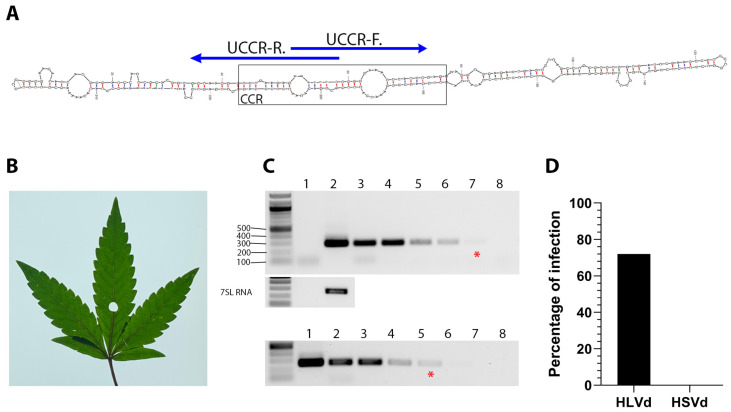
Diagnostic detection of HLVd in infected hemp plants. (**A**) The central conserved region (CCR) is the most conserved secondary structure among viroid species. Specific detection primers were developed based on the HLVd upper strand of the CCR (UCCR) and yield 256 bp amplicons. (**B**) Leaf from HLVd-infected hemp used for total RNA extraction. (**C**) Ten-fold serial dilutions of RNA extract from HLVd-infected leaves (lanes 2–8, upper panel) and plasmid cDNA copy of HLVd (lanes 1–7, lower panel) were used in RT-PCR and PCR reactions, respectively. The red (*) indicates the lane with the minimal dilution that could be visually detected. Nuclease-free water was used for mock (lane 1 of RNA gel, and lane 8 of DNA gel) treatment. A 1 kb ladder is on the left of each gel and the sizes of several ladder bands are also indicated on the left of the top gel. RT-PCR amplification of the 7SL RNA shown below target band as internal control. (**D**) Survey of HLVd and hop stunt viroid (HSVd) in 111 hemp samples obtained from a hemp nursery.

**Figure 2 viruses-16-00030-f002:**
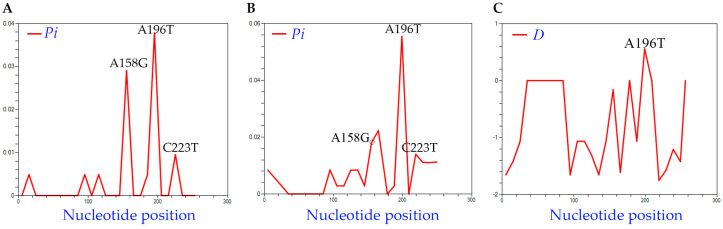
Distribution of nt diversity featuring informative nt positions 158, 196, and 223 in each histogram. The average nucleotide diversity per nt site (*Pi*) (**A**) among the 50 new HLVd isolates recovered in this study and (**B**) among the 77 HLVd genomes used for phylogenetic analysis in Figure 3. (**C**) Distribution of Tajima’s *D* values.

**Figure 3 viruses-16-00030-f003:**
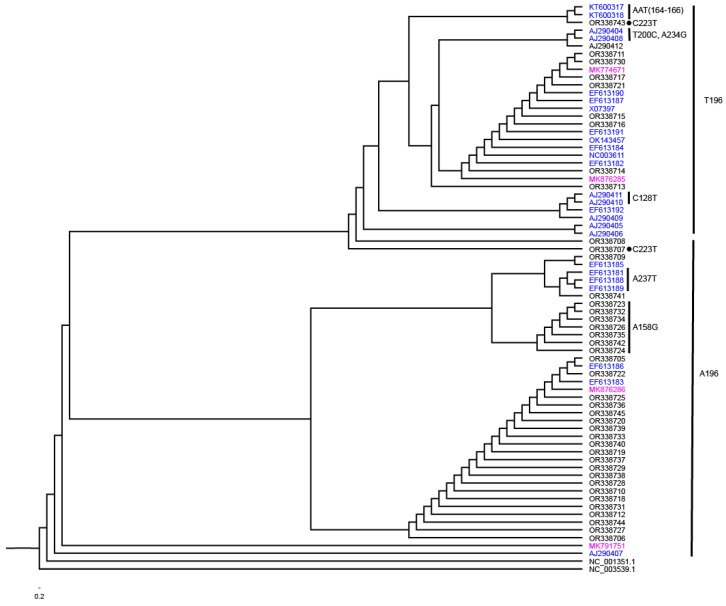
Phylogenetic relationships of HLVd isolates inferred using the maximum likelihood method (1000 replicates). Branches leading to significant clustering of isolates were assigned as a group based on informative nt changes to assist in the explanation of the results. HLVd isolates reported to NCBI by other researchers that are from hops are identified in blue and from hemp are identified in magenta. All other HLVd isolates are reported in this study.

**Figure 4 viruses-16-00030-f004:**
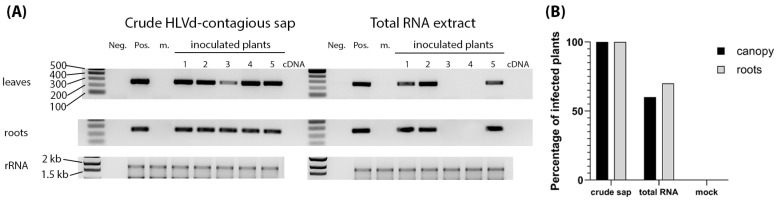
Infectivity of HLVd-containing sap and RNA inoculum. (**A**) Examples of RT-PCR amplicons obtained from inoculated plants at 3 weeks post inoculation. Lanes presenting RT-PCR products representing the results of crude sap and total RNA inoculum are indicated at the top. Samples from individual plants are numbered at the top. Neg. and Pos. indicate the controls. m is a mock control sample. Gels are labeled on the left to identify ethidium-bromide-stained gels with representative RT-PCR amplicons from leaves and root samples, as well as the stained ribosomal RNA (rRNA) presented at the bottom to show sample concentrations were consistent across lanes. The 1kb ladder is on the left of the set of negative and positive controls and the sizes of several ladder bands are also indicated on the left. (**B**) Percentages of total numbers of infected plants pooled together from five sequential trials with 10 plants each.

**Figure 5 viruses-16-00030-f005:**
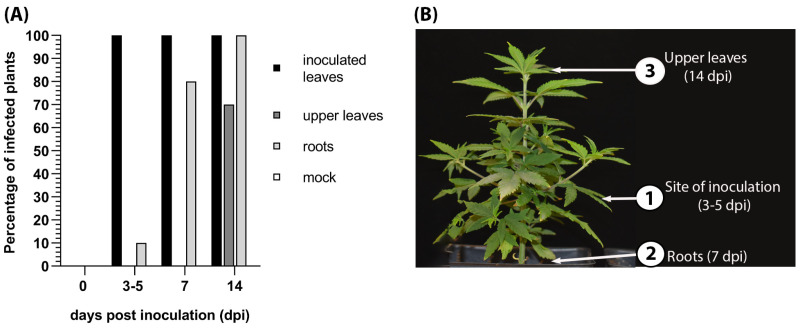
Spatiotemporal distribution of hop latent viroid (HLVd) in infected hemp. (**A**) Graph representing the percent of inoculated leaves, upper leaves, and roots that were positive for HLVd using a RT-PCR diagnostic test. There were zero plants infected with HLVd among the “mock” samples. (**B**) Schematic diagram of HLVd progression upwards in hemp plants showing the sampling locations (1), (2), and (3).

**Figure 6 viruses-16-00030-f006:**
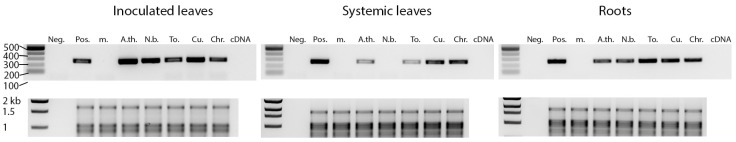
Pathogenicity of HLVd on herbaceous hosts. Freshly prepared inoculum of HLVd was applied to *Arabidopsis thaliana* Col-0 (A.th.), *Nicotiana benthamiana* (N.b.), tomato cv. Moneymaker (To.), cucumber cv. Harris (Cu.), and *Chrysanthemum morifolium* (Chr.). Crude sap from healthy hemp was used for mock control (m.), and nanopure water was used for RT-PCR negative control (Neg.). The expected band size is 256 bp. HLVd detection was performed on the inoculated leaves (left panel), juvenile leaves (middle panel), and roots (right panel). rRNA gels are included below each RT-PCR panel. Sizes of DNA marker bands are indicated next to each band.

**Figure 7 viruses-16-00030-f007:**
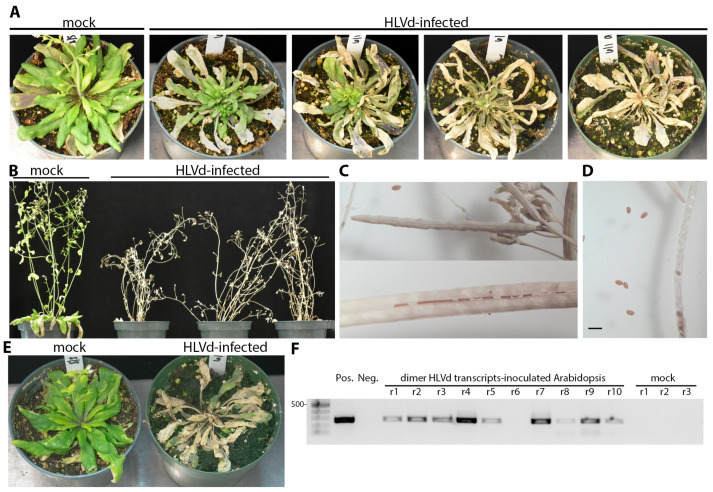
Die-back and tissue collapse of *A. thaliana* (Col-0) upon inoculation with HLVd at two different growth stages. (**A**) Plants inoculated at the growth stage 1.10 of were imaged between 14 and 19 dpi. (**B**) HLVd-infected *A. thaliana* (Col-0) inoculated at the growth stage 1.14 and photographed at 21 dpi. (**C**) Closed (upper) and opened (lower) siliques harvested from infected plants in panel B after dieback showing seeds production appears normal. (**D**) *A. thaliana* (Col-0) seeds collected from HLVd-infected plants. Scale bar equals 2 mm. (**E**) Plants inoculated with infectious HLVd transcripts also produced die-back disease between 14 and 19 dpi. (**F**) Gel image on right shows RT-PCR products from ten plants inoculated with infectious transcripts with nine testing positive for infection. The size marker is the same as in Figure 6 and the 500 bp band is identified on the left side. Plasmid DNA was used for PCR amplification of HLVd coding sequences (Pos.) and nanopure water was used for RT-PCR negative control (Neg).

**Figure 8 viruses-16-00030-f008:**
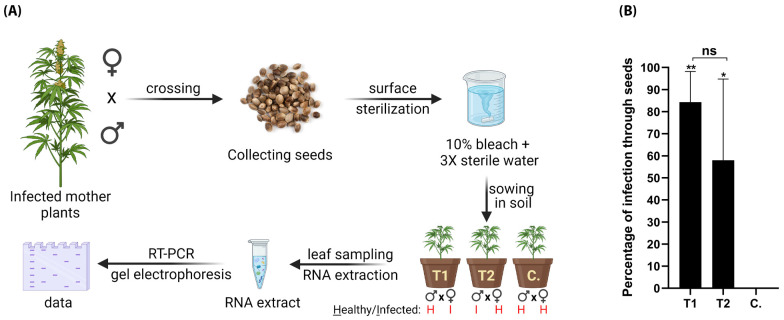
Vertical transmission of HLVd via hemp seeds. (**A**) Schematic diagram illustrating procedures for testing seed transmission. HLVd-infected hemp females crossed with pollen from healthy male hemp (T1), and healthy females crossed with HLVd-infected pollen (T2). Newly emerging plants were RT-PCR tested for HLVd. (**B**) Percentage of HLVd-positive plants emerged from 300 seeds in each treatment and a total of 100 seeds for the control (C.). Results were statistically analyzed using one-way ANOVA followed by Tukey’s post hoc HSD test. Values are the mean of three independent replicates (100 seeds/replicate) and vertical bars are the standard error means. Asterisks (** and *) denote significance between treatments and untreated control at *p* = 0.05. ns: not significant.

**Table 1 viruses-16-00030-t001:** Primers for viroid detection and RNA normalization.

Target	Primer	Primer Sequence 5′-3′	Amplicon Size (bp)	Reference
HLVd	UCCR-F	GGGATCCCCGGGGAAACCTACTCG	256	This study
	UCCR-R	GGGATCCCTCTTCGAGCCCTTGCC		
HSVd	F1	GGGGCAACTCTTCTCAGAATCC	302	[58]
	R1	GGGGCTCCTTTCTCAGGTAAGTC		
7SL RNA	alpha	TGTAACCCAAGTGGGGG	231	[60]
	anti-beta	GCACCGGCCCGTTATCC		

**Table 2 viruses-16-00030-t002:** Infectivity of HLVd infection on *A. thaliana* Col-0, *N. benthamiana*, tomato cv. Moneymaker, cucumber cv. Harris, and *Chrysanthemum morifolium*. Ten healthy plants were used per treatment. HLVd infection was detected using RT-PCR.

	Plant Family:	RT-PCR Detection in:			Symptoms
		Inoculated Leaves	Roots	Upper Leaves	
Negative control	-	0	0	0	Asymptomatic
Arabidopsis	(Brassicaceae)	100	100	70	Die-back, gradual decline
*N. benthamiana*	(Solanaceae)	100	60	0	Asymptomatic
Tomato	(Solanaceae)	100	100	80	Asymptomatic
Cucumber	(Cucurbitaceae)	100	100	100	Asymptomatic
Chrysanthemum	(Asteraceae)	60	40	40	Asymptomatic

## Data Availability

Sequences for HLVd isolates are available in NCBI Genbank database.

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
