# Peer review of "New Insights into Hop Latent Viroid Detection, Infectivity, Host Range, and Transmission"

_viruses, 2023, doi:10.3390/v16010030_

Round 1

Reviewer 1 Report

Comments and Suggestions for Authors

Author Response

Comment 1:  All edits provided in the PDF file were incorporated as suggested. 

Comment 2:  The authors used total RNA, not transcript when the authors tested the sensitivity of RT-PCR for HLVd. Are there any specific reasons?

Our main goal was to develop an assay for rapid and accurate detection of HLVd for hemp growers and breeders. We are planning to take this assay forward and make it more adaptable to field situations. Therefore, we assayed primer sensitivity on RNA to make it more comparable for detection in the field.

Reviewer 2 Report

Comments and Suggestions for Authors

The manuscript “New insights into Hop latent viroid detection, infectivity, host 2 range, and transmission” deals with HLVd detection, transmission, and host range. The article also includes the diagnosis of HLVd and HSVd in 111 plants and the molecular characterization of the HLVd population in some positive samples.

The paper is well written, the study design is adequate, figures are clear, subject, results and discussion are appropriated for “ Viruses”. However, I have a major criticism about this work and that is that it has been performed using exclusively RT-PCR as a diagnostic method. RT-PCR for the detection of viroids involves risks of false positives and false negatives, even more so when a high number of cycles is used, as in the case of this work where 35 cycles are applied.

The authors have added a good number of controls to their assays and the results of their analyses correlate very well as they do not find infected samples in them. However there are two points that I consider should be supported by a second diagnostic method (molecular hybridization or NGS) given their relevance.

The first point is the infection of Arabidopsis thaliana by HLVd. Many groups have tried to infect this species with viroids and none have succeeded. The closest approach in this regard has been the work of Daros and Flores 2004 (DOI: 10.1073/pnas.0401090101) where viroid sequences were transgenically inserted to prove that Arabidopsis thaliana has the necessary machinery for viroid processing. Proving that this model plant can be infected with viroids is very relevant and therefore I consider that this claim does have to be supported by a second diagnostic method (NGS, hybridization). The authors indicate that HLVd infection induces symptoms but it must be taken into account that the inoculum used to produce the infection was a crude extract of infected plants and therefore infection by a pathogen other than HLVd cannot be ruled out. NGS analysis or inoculation of HLVd by synthetic dimeric transcripts of this viroid would be necessary for HLVd with the symptoms shown by the plants.

The second point is vertical transmission. This article indicates a transmission rate of 84.3% (infected female parent) and 58% (infected male parent).  There are few precedents of such a high vertical transmission of viroids and the repercussion of this result in the Cannabis cultivation sector is very great. Therefore, I believe that analyzing this experiment with a second diagnostic method such as dot-blot molecular hybridization would be very convenient.

Minor consideration:

This work performs HSVd diagnosis using HSVd F1 and HSVd R1 primers described in (Bernad and Duran-Vila 2006) and designed to detect HSVd in citrus. HSVd primer covers at the 3' end position 59 of HSVd and this position has a host-dependent variability described by Sano and Kashiwagi (2020, DOI: 10.1016/j.virusres.2022.198706). If HSVd in Cannabis varies at that position it would give a negative diagnostic result. I feel that this information would need to be reflected in the discussion.

Author Response

Comment 1:  RT-PCR for the detection of viroids involves risks of false positives and false negatives, even more so when a high number of cycles is used, as in the case of this work where 35 cycles are applied.   We used RT-PCR detection which is the most common technology across viroid research.  Our data shows in Figure 1, Supplementary Figure 1 the sensitivity and specificity of the primers.  We saw amplicons for HLVd after 25 to 30 PCR cycles but we reported using 35 cycles in this study to feature the sensitivity of our primers, detecting sub-femtograms of viroid.   False negative results are typically due to inhibitors in the extract and these inhibitors can be overcome in a dilution series typically.  We use the Promega Maxwell robot for all RNA extractions for diagnostics in a wide range of studies.  We found over the past 10 years that the quality of RNA is better than triazol and Qiagen and there are less inhibitors in our extracts.  The quality of RNA using this extraction method has been published extensively by our team and comparison with other technologies was reported by the Verchot laboratory in Chuahan et al., 2015.  We have run RNA quality analysis and the quality is good enough for RNA_seq which is unlike triazol or organic extraction technologies.  If there were inhibitors in the RNA preventing detection of viroid RNA, then there would be less product at high concentration of RNA and more products at lower concentrations.  Figure 1 shows that with increasing dilution of total RNA, there is a similar linear reduction in PCR products across dilutions.  Supplementary Figure 1 presents control experiments run to test for false-positives.  False positives would occur from self-priming of viroid RNA working with the RT in the first step reaction.  In this case we did not see false priming.  The supplementary data shows that priming is specific using our designed primers.  We included cDNA controls, no-cDNA controls, etc.  In Supplementary Figure 2 we sent all PCR products for sequencing to verify there are HLVd specific bands.  Many of the samples were also tested by a third party and we found that our assay was more sensitive and their assays produced false negatives, probably because of the way they extracted their RNA which did not eliminate inhibitors. 

Comment 2:  There are two points that I consider should be supported by a second diagnostic method (molecular hybridization or NGS) given their relevance.  The first point is the infection of Arabidopsis thaliana by HLVd. Many groups have tried to infect this species with viroids and none have succeeded.

We reviewed the literature and there is only one published study by Daros and Flores 2004 (DOI: 10.1073/pnas.0401090101) testing viroids in Arabidopsis where viroid sequences were transgenically expressed, proving that Arabidopsis thaliana has the necessary machinery for viroid processing. According to a review article by Kovalskaya and Hammond 2014, there are no reports of Arabidopsis infection by viroids, but suggesting it is not a “natural host” for most viroids, however the absence of data is not evidence that it cannot occur in the laboratory.  Notably in Solovyev et al they reported the possible role of Nt-4/1 protein in systemic viroid transport noting that Arabidopsis has a homologue that is known to contribute to virus movement. 

The reviewer asks us to consider that the sap inoculum used in the study might be contaminated with another virus that we did not identify.  Therefore, we added panels E and F to figure 7 showing the die-back disease produced by using in vitro transcripts of HLVd which is the same as when we used sap.  So the outcomes of HLVd inoculation by sap or in vitro transcripts are the same. We prepared added explanation of the infectious clone to the M&M.  

Finally, we performed a dot blot which we are attaching here to satisfy the reviewer, but do not intend to include as a supplementary figure to the paper, unless the reviewer requires this.  In this blot we took RNA from 6 Arabidopsis plants (all infected), 6 hemp plants (4 infected and 2 negative by RT-PCR), and a series of controls inlcuded infected hemp, and healthy/mock controls from both plants.  The blot shows accumulation of plus and minus strand RNA which is consistent with accumulation of a replicating viroid. 

Comment 3:  This article indicates a transmission rate of 84.3% (infected female parent) and 58% (infected male parent). There are few precedents of such a high vertical transmission of viroids and the repercussion of this result in the Cannabis cultivation sector is very great. Therefore, I believe that analyzing this experiment with a second diagnostic method such as dot-blot molecular hybridization would be very convenient.

HLVD is a disease of the flower (dudding disease) so it is not surprising that it would get into the seed.   Since these hemp plants are mechanically crossed there is a higher than naturally occurring % of infection.  This may be an artificial system because crosses were performed in doors.  A lower rate is reported in hops grown in open fields where plants were not crossed to test for infection.  Other reports are natural occurrences in seeds of plants. This means that breeders cannot work with infected material and expect to produce clean material.   We also used a third party to confirm these results but do not have any agreements in place to report their data in any publication.   Growers are reporting on their websites high seed transmission.  I am aware the implications of this for the industry. 

Comment 4:  This work performs HSVd diagnosis using HSVd F1 and HSVd R1 primers described in (Bernad and Duran-Vila 2006) and designed to detect HSVd in citrus. HSVd primer covers at the 3' end position 59 of HSVd and this position has a host-dependent variability described by Sano and Kashiwagi (2020, DOI: 10.1016/j.virusres.2022.198706). If HSVd in Cannabis varies at that position it would give a negative diagnostic result. I feel that this information would need to be reflected in the discussion.

Please note: I have tried several times to upload the blot.  for some reason the coverletter attached below and then a Report notes was added.  I hope that you can see all the informaiton that I provided embedded in this response and attached.  

Reviewer 3 Report

Comments and Suggestions for Authors

The manuscript viruses-2718768 entitled » New insights into Hop latent viroid detection, infectivity, host range, and transmission« is scientifically set, and adequately prepared according to journal guidelines. The manuscript presents important new research findings on HLVd. Before publishing I recommend few corrections. Please find below comments on the manuscript:

Page 1, Line 35: I am not familiar with references where it is reported that spider mites can transmit viroids. The references that are cited do not provide evidence for that. I suggest that you cite following article Hadidi A, Sun L, Randles JW. Modes of Viroid Transmission. Cells. 2022 Feb 18;11(4):719. doi: 10.3390/cells11040719. PMID: 35203368; PMCID: PMC8870041.

Page 2, Lines 70-72: Include reports for HLVd presence for Czech Republic, Slovenia, USA, and China.

Look: Patzak J, Henychová A, Krofta K, Svoboda P, Malířová I. The Influence of Hop Latent Viroid (HLVd) Infection on Gene Expression and Secondary Metabolite Contents in Hop (Humulus lupulus L.) Glandular Trichomes. Plants (Basel). 2021 Oct 26;10(11):2297. doi: 10.3390/plants10112297. PMID: 34834660; PMCID: PMC8617911.

Radišek, S.; Oset, M.; Cerenak, A.; Jakše, J.; Knapic, V.; Matoušek, J.; Javornik, B. Research activities focused on hop viroid diseases in Slovenia. In Proceedings of the Scientific Commission of IHGC, Kiev, Ukraine, 4–9 June 2013; pp. 58–3785.

Radišek, S. Management of hop viroid diseases in Slovenia. In Proceedings of the 56th International Hop Growers’ Convention (IHGC) Congress, Yakima,WA, USA, 30 July–3 August 2017.

Stajner, N.; Radisek, S.; Mishra, A.K.; Nath, V.S.; Matoušek, J.; Jakše, J. Evaluation of Disease Severity and Global 626 Transcriptome Response Induced by Citrus Bark Cracking Viroid, Hop Latent Viroid, and Their Co-Infection in 627 Hop (Humulus Lupulus L.). Int J Mol Sci 2019, 20, doi:10.3390/ijms20133154.

Pethybridge, S.J.; Hay, F.S.; Barbara, D.J.; Eastwell, K.C.; Wilson, C.R. Viruses and Viroids Infecting Hop: Signif-652 icance, Epidemiology, and Management. Plant Dis 2008, 92, 324–338, doi:10.1094/PDIS-92-3-0324.

Page 3, Lines 116-117: Add which variety of chrysanthemum you used.

Page 3, Lines 119-122: It is unclear how many plants were sampled. How many samples were from leaves and how many from roots. Is it possible that some leaf and root samples originated from the same plant? Are all the samples sourced from a single nursery? Were all the samples taken in a single year? Do you have information about hemp varieties? Please provide clarification on this matter and include as much information as possible.  

Page 3, Line 144: Add description how did you develop PCR primers for HLVd (software, conditions…)

Page 4, Line 157: Did you use sap directly in an RT-PCR reaction, or did you perform RNA isolation from sap?

Page 6, Lines 266-267: HSVd is not Cocadviroid.

Page 7, Line 277, Figure 3: Discuss the variability of hop and hemp isolates. Mark (Figure 3) which sequences belong to hemp and which to hop.

Page 7, Line 293-294: I don’t see that from Figure4A.

Page 8, Lines 314-320: This explanation is confusing and not in order with Figure 5. Was sampling conducted at 5 dpi or not? If so, please add the data to Figure 5. All additional data that you are explaining, such as negative HLVd leaflets of the inoculated leaf and the leaf directly above the inoculated leaf, cannot be referred to in Figure 5B. Rewrite more clearly.

Page 12. Lines 460-462: Discuss more about HLVd variation obtained in your analysis (hop variants, hemp cultivars….)

Author Response

Comment 1:  Page 1, Line 35: I am not familiar with references where it is reported that spider mites can transmit viroids. Please add reference for Hadidi 2022. 

We made the suggested changes, deleting spider mites and adding the reference by replacing Wei 2019 with Hadidi 2022 (#6). 

Comment 2:   Page 2, Lines 70-72: Include reports for HLVd presence for Czech Republic, Slovenia, USA, and China.  All references are included, and the names of countries added as suggested.

Comment 3:  Page 3, Lines 116-117: Add which variety of chrysanthemum you used.

Added variety Yellow Fall. 

Comment 4:  Page 3, Lines 119-122: It is unclear how many plants were sampled. How many samples were from leaves and how many from roots.   

The number of samples are now provided.   We deleted the statement of “roots” which were included among the samples but not used in this study directly. 

Page 3, Line 144: Add description how did you develop PCR primers for HLVd (software, conditions…)

Description of PCR primers, conditions, and the software was added as requested.

Page 4, Line 157: Did you use sap directly in an RT-PCR reaction, or did you perform RNA isolation from sap?

The sap was subjected to total RNA extraction and RT-PCR, which we added to the text.

Page 6, Lines 266-267: HSVd is not Cocadviroid.

Thank you for catching that, HSVd is a Hostuviroid, and the error was fixed.

Page 7, Line 277, Figure 3: Discuss the variability of hop and hemp isolates. Mark (Figure 3) which sequences belong to hemp and which to hop.

We highlighted HLVd isolates with different colors in Figure 3 based on origin, either hemp or hop. We further discussed the sequence of divergence among those groups of isolates and drew a correlation to the host. We added our explanations to the results section and we discussion our findings in the discussion section.

Page 7, Line 293-294: I don’t see that from Figure4A.

Text was corrected to match the outcomes from Figure 4A.

Page 8, Lines 314-320: This explanation is confusing and not in order with Figure 5. Was sampling conducted at 5 dpi or not? If so, please add the data to Figure 5. All additional data that you are explaining, such as negative HLVd leaflets of the inoculated leaf and the leaf directly above the inoculated leaf, cannot be referred to in Figure 5B. Rewrite more clearly.

The entire paragraph was rewritten in a clearer and more direct way, so that it describes the experimental steps and matches the results shown in Figure 5. Sites of inoculation were added to the text.

Page 12. Lines 460-462: Discuss more about HLVd variation obtained in your analysis (hop variants, hemp cultivars….)

Sequence variation among HLVd isolates derived from hemp and hop was discussion, as explained in the point number 8.

Round 2

Reviewer 2 Report

Comments and Suggestions for Authors

After reading the responses of the authors and having done a hybridization to check by a second diagnostic method the infection in Arabidopsis thaliana I consider that this article should be accepted for publication in its present form.

Author Response

All corrections requested are provided and are highlighted in the uploaded document